# Design of an Integrated Microfluidic Paper-Based Chip and Inspection Machine for the Detection of Mercury in Food with Silver Nanoparticles

**DOI:** 10.3390/bios11120491

**Published:** 2021-11-30

**Authors:** Lung-Ming Fu, Ming-Kuei Shih, Chang-Wei Hsieh, Wei-Jhong Ju, You-Lin Tain, Kuan-Chen Cheng, Jia-Hong Hsu, Yu-Wei Chen, Chih-Yao Hou

**Affiliations:** 1Department of Engineering Science, National Cheng Kung University, Tainan 701401, Taiwan; loudyfu@mail.ncku.edu.tw (L.-M.F.); david6767@gmail.com (W.-J.J.); 2Graduate Institute of Materials Engineering, National Pingtung University of Science and Technology, Pingtung 912301, Taiwan; 3Graduate Institute of Food Culture and Innovation, National Kaohsiung University of Hospitality and Tourism, Kaohsiung 812301, Taiwan; mkshih@mail.nkuht.edu.tw; 4Department of Food Science and Biotechnology, National Chung Hsing University, Taichung 402202, Taiwan; welson@nchu.edu.tw; 5Department of Medical Research, China Medical University Hospital, Taichung 404328, Taiwan; 6Department of Pediatrics, Kaohsiung Chang Gung Memorial Hospital and Chang Gung University College of Medicine, Kaohsiung 833401, Taiwan; tainyl@hotmail.com; 7Institute for Translational Research in Biomedicine, Kaohsiung Chang Gung Memorial Hospital and Chang Gung University College of Medicine, Kaohsiung 833401, Taiwan; 8Institute of Food Science and Technology, College of Bioresources and Agriculture, National Taiwan University, Taipei 106216, Taiwan; kccheng@ntu.edu.tw; 9Institute of Biotechnology, College of Bioresources and Agriculture, National Taiwan University, Taipei 106216, Taiwan; 10Department of Medical Research, China Medical University Hospital, China Medical University, 91 Hsueh-Shih Rd., Taichung 404328, Taiwan; 11Department of Optometry, Asia University, Taichung 413305, Taiwan; 12Department of Seafood Science, National Kaohsiung University of Science and Technology, Kaohsiung 811213, Taiwan; F107176106@nkust.edu.tw; 13Department of Medicine, Chang Gung University, Linkow 333423, Taiwan; naosa720928@gmail.com

**Keywords:** microfluidic paper-based analytical device (μPAD), silver nanoparticles (AgNPs), mercury (Hg), RGB

## Abstract

For most of the fast screening test papers for detecting Hg^2+^, the obtained results are qualitative. This study developed an operation for the μPAD and combined it with the chemical colorimetric method. Silver nanoparticle (AgNP) colloids were adopted as the reactive color reagent to combine and react with the Hg standards on the paper-based chip. Then, the RGB values for the color change were used to establish the standard curve (R^2^ > 0.99). Subsequently, this detection system was employed for the detection tests of actual samples, and the detected RGB values of the samples were substituted back to the formula to calculate the Hg^2+^ contents in the food. In this study, the Hg^2+^ content and recovery rate in commercially available packaged water and edible salts were measured. The research results indicate that a swift, economical, and simple detection method for Hg^2+^ content in food has been successfully developed.

## 1. Introduction

The basic requirement and prerequisite for maintaining physical and mental health is to consume safe and nutritious food [1]. However, consistently increasing human activities and rapid industrialization have been releasing excessive amounts of pollutants to the environment. Water pollution caused by heavy metals such as metallic mercury (Hg) has become one of the most serious pollution issues worldwide. Because minute amounts of Hg can cause damage to the nervous, urinary, reproductive, immune, cardiovascular, and endocrine systems, mercury pollution of natural water bodies exposes human health and ecosystems to severe risks [2,3]. Mercury ions (Hg^2+^) are often adopted as a substitute for inorganic mercury (Hg^2^*^+^*), which is one of the most dangerous environmental pollutants. Hg^2+^ is the most stable oxidation state of mercury, and owing to its soluble form, it is more likely to pollute water resources [4,5,6]. Therefore, it is crucial to determine the Hg^2+^ content in water samples. As is well known, techniques for highly sensitive and selective detection of Hg^2+^ include cold vapor atomic absorption spectrometry (CVAAS), cold vapor atomic fluorescence spectrometry (CVAFS), and inductively coupled plasma mass spectrometry (ICP-MS). However, these high-performance techniques often require expensive equipment and technical expertise for sample preparations [7,8,9,10]. 

As stated by the World Health Organization (WHO) in the ASURED Challenge [11], access to equipment should not be a limitation to diagnostic test performance, especially for developing countries and other resource-limited regions of the world. The qualitative and quantitative detections of all types of chemical hazards require an approach that is fast, simple, and inexpensive. Accordingly, the microfluidic paper-based analysis device (μPAD) [12,13] has garnered widespread attention as a promising candidate device since its introduction in 2007 [12]. In the past decade, a significant number of research papers have been published in the μPAD field [14]. Several publications on the μPAD were focused on the detection of hazards in food or novel manufacturing methods and designs, as well as proof-of-concept applications for detecting chemical contaminants that could be present in food. The μPAD adopts a filter paper as the substrate for the chip, which is then processed with different physical and chemical methods, such as cutting, modification, stacking, and coating, to fabricate the chip for detection [12,15]. Its primary concept is to transfer the entire experiment to a palm-sized chip for testing, thereby reducing the amount of materials consumed in the experiment and minimizing the cost, such that the idea of lab-on-a-chip (LOC) is realized [16]. Microfluidic devices exhibit the potential to perform high-throughput and high-efficiency biochemical detection experiments that consume minimal amounts of samples and reagents, with low detection limits. In addition to other advantages, microfluidic devices and chip detection experiments exhibit higher economic efficiency, higher portability, and simpler operations than conventional detection methods [17,18,19,20]. 

The fabrication methods for paper-based chips include wax printing, photolithography, laser cutting, hydrophobic material coating, inkjet printing of permanent marker ink, silkscreen printing, colloidal precipitation coating, and permanent glue coating [21]. The filter paper has a porous fiber structure and exhibits good capillary water absorption properties. Therefore, parameters such as the amounts of reagents required for the reaction, including the diffusion path and area of the liquid, can be configured via processing technologies. In addition, it is biodegradable and can alleviate waste accumulation and pollution. Wax printing is a technique that applies wax with hydrophobic properties on the filter paper surface; the wax is then melted and penetrates the filter paper fibers via heating and baking, thereby covering the filter paper fibers and preventing the solution from being absorbed, such that the hydrophobic effect is achieved. A detection pattern is then designed to create the hydrophobic and hydrophilic areas, such that the reaction solution in the hydrophilic area is not diffuse, and the color reaction can be concentrated without being dragged away by the capillary effect of the filter paper [22,23]. The principle of photolithography-based chip fabrication is similar to that of wax printing, except that, in this method, the coating medium on the filter paper is a hydrophobic polymer, which is cured on the filter paper substrate by light; similarly, hydrophobic and hydrophilic areas can also be created to allow the color reaction to proceed [12]. Various processing methods exist for silkscreen printing. One method is to coat light-curing reagents on a metal silkscreen, after which a filter paper is pressed under the pre-treated metal silkscreen; a hydrophobic medium (such as wax) is then applied, which passes through the metal mesh plate and attaches to the filter paper. The other method is to first cut a designed pattern and then coat it on the filter paper using a hydrophobic medium. In both methods, the hydrophobic media need to be further processed to create hydrophobic and hydrophilic areas to facilitate detection [24,25]. 

Colorimetry is a commonly used analytical technique that does not require complicated equipment design and implementation. This method is based on monitoring color changes in solutions, such as adopting ultraviolet–visible (UV–Vis) absorption spectroscopy that relates to the analyte concentration. The colorimetric method was originally introduced by Birch and Stickle [26]. The colorimetric device comprises a scanner that generates light and an image processor based on the RGB color system, where the light intensity is converted into absorbance units [27,28,29]. Ag particles can combine with mercury to form a Hg-Ag alloy, during which Ag primarily undergoes an oxidation–reduction reaction. The reduced form of silver nanoparticles (AgNPs) will exhibit a color reaction with Hg^2+^, and its original yellow color will become colorless. Therefore, by utilizing the color reaction between AgNPs and Hg atoms, a stable, accurate, sensitive, and simple colorimetric interpretation method can be developed, which will facilitate the real-time detection of mercury pollution. 

AgNPs can be produced using physical, chemical, green chemical, and optical methods, as well as other methods [30,31,32]. Among them, the chemical reduction method is relatively more common, which can produce AgNPs easily. In this method, there are three reactants, which include the Ag precursor, coating agent, and reducing agent. The Ag precursor is reduced to the particle state by the reducing agent. Subsequently, the outer layer of the Ag particles is coated by the coating agent to prevent agglomeration. The morphologies of Ag particles can also be altered by adjusting the concentration and the added amount of the reaction reagent, as well as other reaction factors. It is well known that the highly sensitive and selective analysis of Hg^2+^ can be realized via the decomposition of unmodified AgNPs, as reported in previous studies [33]. AgNPs can be oxidized by Hg^2+^ and decomposed into even smaller particles and Hg(o), which may also be deposited on the surface of AgNPs, thereby generating Hg(o) amalgam particles (Hg-Ag) [34]. Recently, Meelapsom et al. performed a proof-of-concept experiment for the application of the μPAD for the determination of Hg^2+^ in water via a chromatic analysis based on a simple RGB color model [35]. The reaction mechanism of unmodified AgNPs and Hg^2+^ on the double-layer μPAD can be explained as follows: The AgNPs are oxidized by Hg^2+^ and decomposed into smaller particles, and Hg^2+^ is reduced to Hg(o). Subsequently, Hg(o) is deposited on the surface of AgNPs, thereby forming Hg-Ag particles [34,35]. 

According to the national announcement by the Republic of China, the detection limits of mercury and methyl Hg differ in various foods; for example, the mercury limit for packaged water and edible ice is 1 ppb (parts per billion), and that for salt is 100 ppb. Detection methods in the announcement primarily include ICP-MS, atomic absorption spectrometry (AAS), atomic fluorescence spectroscopy (AFS), and high-performance liquid chromatography (HPLC). As mentioned before, although these advanced precision instruments can deliver substantially accurate results, the corresponding cost burden is also relatively high, making the general manufacturers in the food industry unable to control and analyze test objects regularly. Recently, several researchers have devoted their efforts and expertise to the development of mercury detection in food and the environment, in order to make the detection process more accessible and economical. The main development trend is oriented towards the fast screening test paper method [36]. In this study, the applications of microfluidics and chemical colorimetry were combined, and AgNPs were employed as the color reagent to develop the detection method for mercury in food. The μPAD was developed to detect the mercury content in food, including actual edible salt samples from different origins. Finally, a shelf life test was conducted to simulate the commercialization possibility of this detection system in the future.

## 2. Materials and Methods

### 2.1. Chemical Reagents

Silver nitrate was purchased from Jingming Chemical, Taiwan. Sodium hydroxide, citric acid monohydrate, trisodium citrate dihydrate, and magnesium sulfate anhydrous were purchased from SHOWA, Japan. Sodium borohydride, copper (II) nitrite trihydrate, and sodium nitrite were purchased from ACROS, USA. Glycerol and iron (III) nitrate nonahydrate were purchased from Alfa Aesar, USA. Hg (II) nitrate hydrate and nickel (II) nitrate hexahydrate were purchased from VETEC, USA. Lead (II) nitrate was purchased from J.T. Baker, USA. Nitric acid and hydrochloric acid were purchased from AENCORE, Australia. The mercury standard solution (1000-μg/mL Mercury AA Standard Solution) was purchased from AccuStandard, New Haven, CT, USA.

Edible salts such as Unification of Life sun-dried sea salt (origin: Spain), Taiyen Biotech iodized high-quality salt (origin: Taiwan), and Marsel natural fine sea salt (origin: Belgium) were purchased from Carrefour, Taiwan. Taiyen Biotech healthy iodized sea salt (origin: Taiwan), Unification of Life sun-dried iodized sea salt (origin: Melbourne, Australia), and Natural Salt New Zealand sun-dried natural sea salt (origin: Marlborough, New Zealand) were purchased from PX mart, Taiwan. Kirkland Signature pure sea salt (origin: Australia) was purchased from Costco Wholesale Corporation, Taiwan. 

### 2.2. Paper-Based Chip Design and Fabrication Process

Paper-based chip design. The paper-based chip is an experimental material formed using filter paper and acrylic. Advantec 1 filter paper was selected as the chip substrate for the chemical color reaction. A wax-jet printer (Xerox, ColorQube 8570, Norwalk, CT, USA) was used to perform the wax treatment. Subsequently, a piece of acrylic was processed into a black card using the CO_2_ laser processing machine (M3 Trading, Mercury III, Miami, FL, USA). The two were then combined to form a paper-based chip. The experimental material was designed as illustrated in Appendix A. The drawing software CorelDRAW 12.0 was used to design the pattern for the paper-based chip. First, with a sheet of 9 mm-diameter No. 1 filter paper as proof, the wax-jet printer was used to print wax on the surface of the filter paper, which verified that the pattern design agreed with that of the actual chip. Then, a No. 1 filter paper with a diameter of 185 mm was employed to print the pattern in a larger quantity. The concept of the paper-based chip is to adopt a general qualitative filter paper as the substrate for the chemical color reaction and coat it with special materials. Then, the coated filter paper is further processed to facilitate the appearance of color changes in chemical color reactions on the paper for subsequent testing and detection. In this study, wax printing and baking were employed to fabricate filter paper-based chips. In this heating process, the wax coated on the surface of the filter paper will dissolve and penetrate the filter paper, such that the paper structure forms a barrier and blocks the diffusion of solutions, thereby creating hydrophilic and hydrophobic reaction volumes on the filter paper (Appendix A). Initially, the filter paper’s appearances after wax printing and baking treatment were observed with the naked eye; then, a field scanning electron microscope (FSEM, JEOL, JSM-6330TF, Peabody, MA, USA) was utilized to reveal the structures of the filter paper for the paper-based chip. 

### 2.3. Design and Operation of the Integrated Microfluidic Paper-Based Chip and Inspection Machine

Design and principle of the integrated paper-based chip and inspection machine. The paper-based chip detection method is based on the chemical color reaction. The color changes of the reactants were converted into the three primary colors, R (red), G (green), and B (blue), which were presented in the corresponding values. Then, a standard curve was established for the change in color intensity with the increase in the reactant concentration. Subsequently, the sample was measured, and the obtained RGB values from the sample were substituted back to the standard curve formula to calculate the concentration of the test substance in the sample. The digital camera and image processing software with the Generalized Assorted Pixel [37,38] function can measure the shape and color of the observed image and recognize the difference in image resolution. The main design principle of the inspection machine is to record the RGB values of the chemically colored blocks on the paper-based chip under fixed temperature and light conditions. Additionally, the smartphone camera lens and software operation can be used to measure the resolution gap. Therefore, the machine needs to be designed as a closed box, as illustrated in Appendix A. A simple slot needs to be opened on the machine enclosure to allow the chip to be inserted for detection. Furthermore, an application for color detection should be equipped; a temperature controller and light source stabilizer should be incorporated. Subsequently, the inspection machine can be used to perform detection operations. 

### 2.4. Color Appearance of Agnps on the Paper-Based Chip

The AgNPs were generated by a simple and easy-to-use chemical reduction method using a silver precursor, capping agent, and reducing agent [39]. Preparation of reaction reagents. AgNPs: An appropriate amount of deionized (DI) water was added to a reaction flask, and then the Ag precursor (silver nitrate) and coating agent (trisodium citrate dihydrate) were added and mixed. Subsequently, the reducing agent (sodium borohydride) was introduced to generate the solution of AgNPs. Once the reaction was complete, the solution was stored in the dark at a low temperature. Color reagent for the paper-based chip reaction: A color reaction was performed on the filter paper. Even with the hydrophilic and hydrophobic areas created for detection, the solution was still diffused instantaneously when dropped onto the filter paper, owing to the capillary effect, triggering uneven color distributions on the filter paper chip. Therefore, the reaction reagent was added to a fixed proportion of humectant [40] to delay the capillary phenomenon and allow the color reaction to show the colors completely in the detection area. Here, 900 μL of AgNPs was taken and transferred to a 1.5 mL microcentrifuge tube; then, 100 μL of glycerol was added and mixed well with a shaker for later testing. The formulated color reagent was dropped on the paper-based chip and then freeze dried in a freeze dryer (FD-8510T, Taiwan) for 24 h. Subsequently, to compare the appearance of different ratios of color reagents in the fiber structure of the filter paper, the paper-based chip was removed and sealed in a light-proof container for FSEM observation.

### 2.5. Establishment of *Mercury* Standard Curve with the µPAD

In conventional optical experiments, the fabrication process for AgNPs is stabilized. According to spectroscopy, different volume ratios of the AgNP color reagent and mercury will exhibit different color changes. Spectroscopy measures the absorption spectrum, while the paper-based detection method measures the changes in the RGB values of the reflected light. Hence, the behaviors of colors produced by different concentrations differ from the absorbance values produced by different concentrations. Therefore, in this section of the experiment, it is necessary to identify the optimal testing condition parameters for the paper-based detection method. Specifically, the following aspects are discussed herein, including the testing environment’s light intensity and temperature, the reaction time point, and the optimal volume ratio for the detection. Mercury standard solutions were employed for the detection tests. The detection steps are presented in Appendix A. The color reagent and standard solution were dropped onto the chip, and then the inspection machine was utilized to interpret the RGB values of the color, thereby confirming the optimal testing parameters. 

(1)Experiment on the Testing Environment’s Light Intensity

The testing environment’s light intensity will influence the RGB value judgment during detection. By adjusting the intensity of the light source, the lighting condition that can deliver the optimal detection colors was tested. In the experiment, the light source in the inspection machine used volts (V) to provide the different light intensities for the photographing component. In this test, the RGB values of the actual color of mercury detected by AgNPs were adopted as the reference, and standard color blocks were drawn with CorelDRAW. The RGB values of standard color blocks were detected under four sets of environmental parameters (2.6/2.6 V, 2.7/2.7 V, 2.8/2.8 V, and 2.9/2.9 V), and the resolution gaps were calculated to determine the light intensity parameter for subsequent tests. 

(2)Optimal Reaction Temperature and Time Point for the Detection

When AgNPs that participate in the color reaction react with the mercury standard, an optimal temperature and a time point exist for the detection, which need to be confirmed, especially when using RGB values as the basis for the detection results. In this experiment, three sets of environment temperatures for the machine, 37 °C, 40 °C, and 50 °C, were adopted to test the reactions between AgNPs and the mercury standard solution (2 ppb) with the changes in time and temperature. The changes in RGB values were measured, and the results obtained under different test environment temperatures were compared to determine the optimal temperature and time point for the detection.

(3)Optimal Volume Ratio Between the Color Reagent and the Test Substance for the Color Reaction

The reaction volume ratio between the color reagent and test substance influences the color change result. For this experiment, we followed the procedure in a previous research report [41]. First, the two reagents were mixed at the same volume ratio, which was set as the initial test condition. Subsequently, the volume of the color reagent was either doubled or halved to verify whether a better color result would appear. Therefore, in the optimal volume ratio experiment, the AgNPs and the mercury standard (0.1–2 ppb) were mixed at ratios of 1:0.5, 1:1, and 1:2. The maximum resolution gaps between test results were calculated and compared to determine the ratio that provides the highest resolution. In addition, it should be noted that the color result worsened with the decrease in the volume of the reaction solution on the loading region of the paper-based chip. Therefore, once the detection volume ratio was determined, the solution volumes were multiplied according to the reaction solution ratio, and the test results were compared. 

(4)Establishment of the Mercury Standard Curve

After the optimal testing conditions were confirmed by referencing the detection limits of mercury in different foods provided by the national regulation, the concentration range of the mercury standard curve was established in the range of 0.1–3 ppb, owing to the fact that the concentration of the test substance will be diluted by the use of various reagents during extraction, when inspecting and analyzing the food. With the concentration range established, water quality detection can also be performed directly simultaneously. If the detection result exceeds the concentration range, the solution can also be further diluted and tested again to obtain the result. However, the detection range optimized in this study is for national regulations and standards related to food quality, not to establish the detection limit (LOD) of the μPAD device. However, the detection range (0.1-3ppb) of the standard calibration curve established in this experiment is no less than UV–Vis (0.1ppb) [42] and better than CVAAS (0.25ppb) [43].

(5)Actual Sample Analysis and Shelf life Test

For the detection tests of actual samples, salts purchased from stores and supermarkets were used to determine their mercury contents. The detection limit of mercury in edible salt permitted by the government is 100 ppb. The salt samples were prepared in water solutions, and a known mercury standard was added to prepare a 100 ppb solution. Samples were tested using µPADs, UV–Vis, and CVAAS and then transferred to a third-party inspection unit. The final test results for a total of 13 samples were compared and analyzed. For the storage and effectiveness of the chips, the prepared paper-based chips containing a AgNP color reagent for mercury detection were stored at low temperatures of 4–7 °C for 1–7, 14, 21, and 28 days. Changes in the chips over time were observed, and the mercury standards were adopted for the tests.

## 3. Results and Discussion

### 3.1. Observation of the Structure of AgNP Color Reagent on the Paper-Based Chip

The AgNP color reagent used for the paper-based chip detection was dropped onto the chip, and its morphology on the filter paper fiber was observed under an SEM, as illustrated in Figure 1A. After dropping AgNPs onto the chip, they could be observed as small white dots on the filter paper fibers. This observation confirmed that the AgNPs were attached to the paper-based chip for the detection reaction [44]. Owing to the capillary effect of the filter paper, the color reagent solution spread rapidly and was distributed unevenly. Thus, the humectant glycerol was introduced to delay the capillary phenomenon, thereby allowing the color reagent to disperse evenly on the paper-based chip [40]. To observe the detailed structures of the fibers, SEM images were taken at 100×, and TEM images of silver nanoparticles (AgNPs) were captured, where the measured particle size averaged around 43 nm, as illustrated in Figure 1B,C, respectively. The formation of AgNPs is related to the reaction conditions. The procedure for this experiment can be found in our previous report [41]. The measured particle size was in the range of 20–110 nm, and the size with the highest frequency was approximately 47 nm, similar to the results obtained using a nanoparticle size analyzer [45].

### 3.2. Optimal Testing Parameters for *Mercury* Detection Using the Paper-Based Chip Combined with AgNPs

#### 3.2.1. The Optimal Light Intensity of the Testing Environment and Its Influence on the Detection Performance

When performing the detection, the intensity of the light source in the testing environment influences the detection results and resolution. Here, the light source environment in the inspection machine was adopted and adjusted in the range of 2.6/2.6–2.9/2.9 V (V) to test the relationships using standard color chips. The obtained results are presented in Figure 2A. As the power supply voltage increases, the intensity and total RGB value increase accordingly. Among the four light source intensities, the resolution gap at 2.6/2.6 V was as high as 86; however, the linearity (R2) trend observed at 2.7/2.7 V was better than that at 2.6/2.6 V (2.6/2.6 V: R2 = 0.9985, 2.7/2.7 V: R2 = 0.9987), and the resolution gap also reached 76. We speculate that it may be because the LED brightness is unstable under the condition of 2.6/2.6 V, so the reproducibility of the value is not ideal. Although the value of 2.7/2.7 V is 10 resolution gaps lower than that of 2.6/2.6 V, it also reaches 76, which is an excellent detection level. Hence, 2.7/2.7 V was selected as the optimal testing parameter for the light source of the testing environment.

#### 3.2.2. Discussion on the Optimal Detection Time and Testing Environment Temperature

When performing a chemical color reaction, the reaction time influences the detection result. Here, the optimal detection time point and testing environment temperature are discussed. The mercury standard (2 ppb) was detected using AgNPs, and the tests were performed at 37 °C, 40 °C, and 50 °C. The obtained results are presented in Figure 2B. The detection results were obtained after reacting for 0.5–20 min under the three testing temperatures. At 37 °C, the reaction entered a plateau at 7.5 min (blue line in the figure); at 40 °C, the reaction entered a plateau at 7 min (green line in the figure); and at 50 °C, the reaction entered a plateau at 6 min (red line in the figure). The plateau indicates that the reaction between AgNPs and mercury was terminated and the chip dried out completely. In this case, even for different concentrations, no difference in RGB values would be detected. Therefore, it is impossible to ascertain the change in concentration. As illustrated in Figure 2C, from the measurement results of mercury standards (0.1, 2 ppb) on AgNPs for 0.5–6 min at 50 °C, a large resolution gap can be observed at 2.5 min. In the experiment, it was determined that the resolution gap value close to the plateau is not necessarily the best. However, at 37 °C and 40 °C, the optimal resolution gaps are observed later than those at 50 °C. Therefore, 50 °C was selected as the temperature condition for the detection test, and 2.5 min was selected as the detection time point.

#### 3.2.3. Optimal Reaction Reagent Parameter

Metal nanoparticles have unique properties and applications in many fields, attributed to collective dipole oscillations called surface plasmon resonance (SPR) [46]. Therefore, the SPR phenomenon is very suitable for the colorimetric sensing of Hg^2+^ ions. The interaction between the nanoparticles and the analyte changes the intensity of the absorption band and/or the position in the visible spectrum, which can usually be observed with the naked eye [47]. The limitations of these systems are mainly related to poor selectivity, high detection limits of Hg^2+^, complex probe material synthesis, or complex analysis procedures. To discuss the optimal reaction volume ratio, the total volume of the AgNP reagent (R) and the mercury standard (S) dropped onto the test paper must be fixed. The experimental ratios were designed as 1:0.5, 1:1, and 1:2 (μL). The RGB values were detected using the reagent after reacting for 0.5–6 min at 50 °C, and the resolution gaps were compared. 

The AgNP color reagent is bright yellow; however, because only a tiny amount is dropped onto the paper-based chip, the color appearance is not significant. Under the volume ratio of R:S = 1:0.5, the resolution gap obtained at the 2.5 min test time point has 23 grids, as illustrated in Figure 3A; under a volume ratio of 1:1, the resolution gap reached 20 grids, as illustrated in Figure 3B; and under the volume ratio of 1:2, the resolution gap reached 12 grids, as presented in Figure 3C. The results of this experiment indicate that the volume of mercury can influence the resolution gap of the test results. This is primarily because the combination of mercury and AgNPs will change the color from yellow to colorless. In addition, the higher the mercury volume, the lighter the color, thereby resulting in poor detection results. Previous research reports have suggested that, as the adsorption and binding of AgNPs with mercury ions involve a redox reaction, with low mercury concentrations, the AgNP surface is oxidized, the surface plasma resonance magnitude of the AgNPs is reduced, and the detected substance is dominated by amalgam. With high mercury concentrations, the amalgam formed by mercury ions and AgNPs coats the surface of AgNPs, meaning that the surface of the NPs is no longer oxidized, and the measured absorbance values tend to decrease due to the high refractive index [42,48,49,50]. Our previous research reported that linearity is the evaluation criterion for the best volume ratio, and 400 nm absorbance is used as the observation condition. The best volume ratio of reagents is 1:0.5, in order to achieve a linearity of at least R2 > 0.99 [41]. As the standard concentration of mercury increases, the color produced by the reaction between AgNPs and mercury gradually changes from yellow to light yellow and finally becomes transparent through the adsorption of mercury ions into AgNPs to form amalgam crystals [45,51]. The optimal reaction volume ratio between AgNPs and mercury is 1:0.5, which was adopted in the subsequent tests. 

Concurrently, considering the color reaction performance on the paper-based chip, and in order to make the color change more significant for evaluation, the volumes of the color reagent and the mercury standard were increased to 2:1 and 3:1.5 for detection. As illustrated in Figure 3D, at the detection time point of 2.5 min, the resolution at the volume ratio of 2:1 reached a resolution gap of 41 grids, while the resolution gap measured at the volume ratio of 3:1.5 was only 23 grids, as illustrated in Figure 3E. When the detection time was extended to 5 min, the increase in the RGB intensity slowed down (Figure 3A,D). Therefore, we tested the appropriate reaction conditions for 2.5 min. The volume ratio of 2:1 and detection time of 2.5 min were finally adopted to establish the standard curve. The resolution gap values measured at different volume ratios of the color reagent and mercury standard are presented in Appendix A. 

In recent years, the number of efforts focused on developing new nanoparticle sensors for mercury detection has increased, mainly due to the need for low-cost portable devices that can provide fast and reliable analytical responses, thereby facilitating analytical dispersion. Botasini et al. reported that the sample matrix highly affects the response of nanoparticle-based sensors [33]. Due to the adverse effects of ionic strength and the presence of exchangeable ligands [52,53], the developed analytical nanosystem may not be used in actual samples. Therefore, the test conditions of this study are very suitable for the use of portable color rendering devices. Thus, optimizing and establishing a detection method for mercury in silver nanoparticles are very worthy of research.

### 3.3. Establishment of the Standard Curve for *Mercury* Detection

At high mercury concentrations, the amalgam formed by mercury ions and AgNPs coats the surface of AgNPs, such that the surface of the NPs is no longer oxidized, and the measured absorbance values tend to decrease due to the high refractive index [45]. After the previous discussion on the optimal testing parameters, the standard curve of mercury in the concentration range of 0.1–3 ppb was established at the AgNP/Hg volume ratio of 2:1 μL at 50 °C and 2.5 min, under a light source intensity of 2.7/2.7 V, as illustrated in Figure 4A. The R2 of all RGB values exceeded 0.99. In addition, as illustrated in Figure 4B, the R2 of the RGB sum reached over 0.99, and the resolution gap value was 64 grids, which can be applied for the detection and analysis of actual samples. The mercury detection standard curve established in this experiment does not involve the LOD of mercury detection, but for the UV–Vis spectrophotometer [42] and CVAAS [43], the mercury detection limits are approximately 0.1 ppb and 0.25 ppb, respectively. This study’s optimized AgNP color rendering range is about 0.1–3 ppb, with great application potential. Previous studies have reported that RGB can be used for PAD color rendering [35,54].

### 3.4. Analysis of Actual Samples

The recovery rate of the test sample was 70–120% in the concentration range of 0.01–0.1 ppm mercury, as developed in the calculation methodology presented by the Taiwan Food and Drug Administration (Validation Criteria of Food Chemical Inspection Methods) [43]. In this study, to test whether µPADs are suitable for determining food mercury content, commercially available salts (sample sources are provided in the Materials and Methods section) were dissolved in ddH2O, and the mercury standard was added to prepare 100 ppb salt water samples. The samples were tested using three methods (µPADs, UV–Vis, and CVAAS) and then sent to a third-party inspection unit for detection. The test results of the samples after the pre-treatment and dilution, obtained using the three methods and by the third party, are presented in Table 1. The third-party inspection result was adopted as the reference to compare the detection results. Although the µPAD data of S1006 (87.22 ppb) are different from the test data obtained by the third-party certification (174.6), we do not rule out the possibility of sampling errors. Since the detection results of µPADs (87.22 ppb), UV–Vis (90.9 ppb), and CVAAS (110.2 ppb) were relatively similar, we retained the data and used them as a follow-up evaluation for the methodological aspects in the analysis of the difference between µPADs and the third-party certification. On the other hand, the W1000 data part was used to dissolve commercially available salts and ddH2O equipped with mercury standard products. Therefore, in µPADs, UV–Vis, and CVAAS, and then after the third-party inspection, it was shown that it cannot be detected (not detectable, ND). As illustrated in Figure 5, the detection results obtained using µPADs are all within the range of the detection recovery rate of 70–120%, which is in line with the criteria presented in the methodology. Taiwan’s national announcement limits the detection of mercury and methylmercury in different foods; for example, the mercury limit for packaged water and edible ice is 1 ppb, while the limit for salt is 100 ppb. Therefore, this research aimed to develop an appropriate detection module for the application of mercury contamination detection in food.

### 3.5. Shelf Life Test

The design of the shelf life test followed that used in a previous study [55]. The AgNP color reagent was dropped onto paper-based chips and stored at a low temperature of 4–7 °C for 1–7, 14, 21, and 28 days. The mercury standard solutions with concentrations of 0.1–3 ppb were adopted for the detection tests. The detection values were expressed by the linearity of the standard curve, and the changes in the chips during storage were observed. 

The obtained results are presented in Appendix A. There were no significant changes in the appearance of paper-based chips containing color reagents. This is because the color reagent dropped onto the chips contains humectant, which can reduce the capillary phenomenon between the liquid and filter paper fibers and cause the color reagent to be distributed evenly on the chip [40]. The presence of humectant makes it possible to eliminate the error of color inconsistency during detection; simultaneously, it can prevent the reagent from volatilizing and dissipating during the shelf life test. When employing mercury standards for the detection tests, the R^2^ exceeded 0.99 during the initial 1–7 days. Because the storage time was prolonged, the R^2^ exhibited a slight downward trend. The R^2^ measured after storage for 28 days dropped to 0.97. This result indicates that when commercializing the paper-based chips in the future, storage at low temperatures of 4–7 °C for approximately one month can provide a certain reference accuracy. If the storage period needs to be further extended, other preservation reagents could be added to the color reagent for testing.

## 4. Conclusions

This study successfully designed an integrated microfluidic paper-based chip and inspection machine, which was employed to perform the detection tests of actual samples. The contributions of this research are as follows. Mercury was detected via its color reaction with AgNPs on the paper-based chip, and a standard curve of mercury in the concentration range of 0.1–3 ppb was successfully established. The R2 of the RGB value exceeded 0.99, the test conditions were in the Hg concentration range of 0.1-0.3 ppb, the volume ratio of AgNP/Hg was 2: 1 μL, at 50 °C/2.5 min, and the light source intensity was 2.7/2.7 V. The volume ratio of the color reagent: Hg standard of 2:1 was the optimal mix ratio for the detection of Hg content. The paper-based chips used of µPADs can be used for approximately one month after storage at low temperatures of 4–7 °C. Furthermore, when the samples were analyzed, the results were compared with the third-party inspection results, and the detection recovery rate was within the range of 70–120%, which is in accordance with the criteria specified by the Republic of China. In addition, three detection methods were compared, and the detection cost and operation convenience were considered. The µPAD developed in this study for mercury detection can function as an optimal detection solution for the practical application industry.

## Figures and Tables

**Figure 1 biosensors-11-00491-f001:**
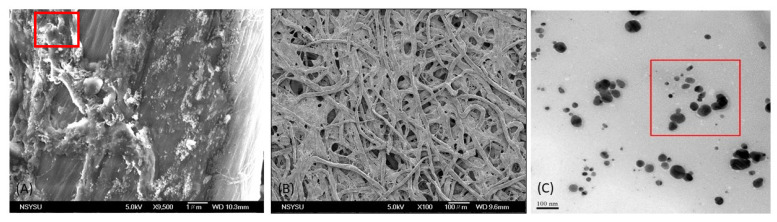
SEM image of paper-based chip: (**A**) silver nanoparticle (AgNP) coating on paper-based chip at 9500×, (**B**) silver nanoparticles (AgNPs) and glycerol observed at 100×, and (**C**) TEM image of silver nanoparticles (AgNPs), where the measured particle size averaged around 43 nm.

**Figure 2 biosensors-11-00491-f002:**
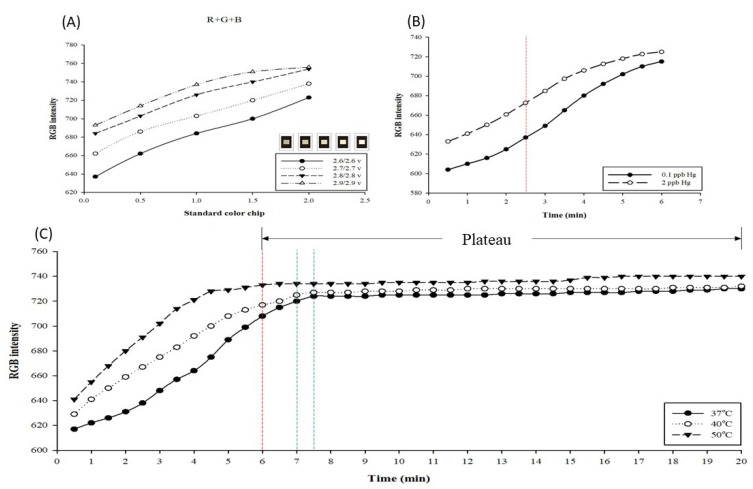
Optimal testing parameters for mercury detection using the paper-based chip combined with AgNPs. (**A**) Determining the three best primary color resolutions of standard color films under different light source intensities, R, G, B, and R+G+B. (**B**) Curve of mercury detection with AgNPs at 37 °C, 40 °C, and 50 °C for 0.5–20 min. (**C**) The curve of the 0.1 and 2 ppb mercury standards for silver nanoparticle detection in 0.5–6 min.

**Figure 3 biosensors-11-00491-f003:**
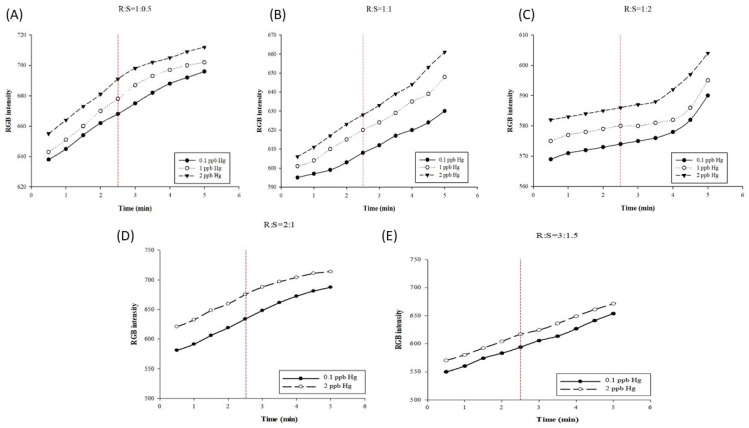
Optimal reaction reagent parameters. Curves of mercury nanoparticle detections in different reaction volume ratios of (**A**) 1:0.5, (**B**) 1:1, and (**C**) 1:3. Curves of mercury nanoparticle detections in different reaction volume ratios of (**D**) 2:1 and (**E**) 3:1.5.

**Figure 4 biosensors-11-00491-f004:**
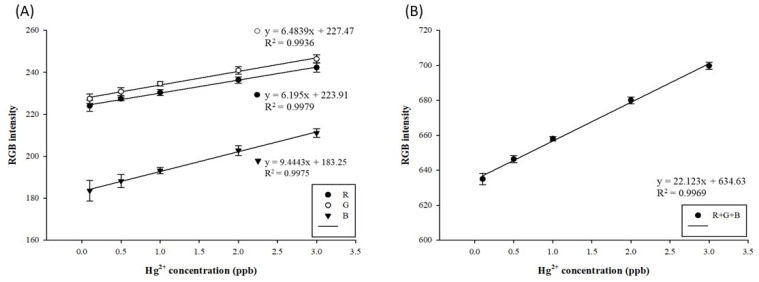
(**A**) RGB and (**B**) R+G+B standard straight curves of paper-based detection systems for mercury by using silver nanoparticles.

**Figure 5 biosensors-11-00491-f005:**
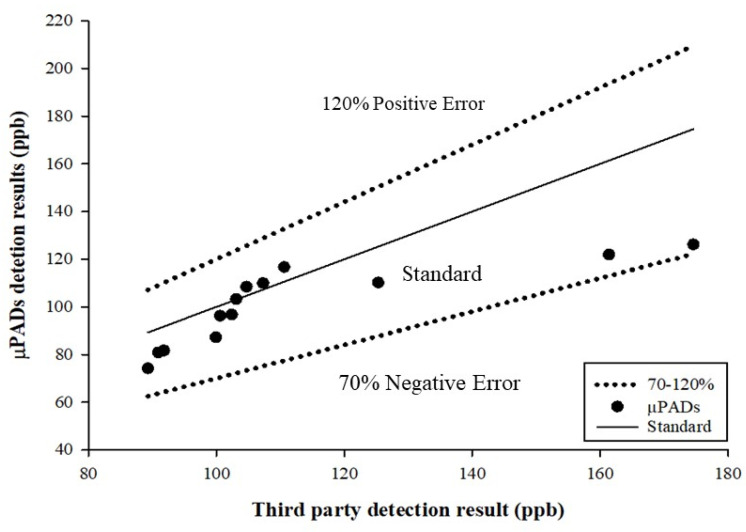
Scatter diagram for the measured sample.

**Table 1 biosensors-11-00491-t001:** Mercury assay results in comparison of different methods, using μPADs, UV–Vis spectrum, CVAAS, and third-party corporation for determination of mercury content.

Mercury Assay (ppb)				
* Sample No.	μPADs	UV–Vis	CVAAS	Third-Party Assay
S1001	126.20	101.2	98.6	110.6
S1002	121.91	90.6	100.1	99.9
S1003	108.46	112.1	99.7	104.7
S1004	109.93	99.2	104.1	107.3
S1005	96.71	87.6	100.8	103.1
S1006	87.22	90.9	110.2	174.6
S1007	80.89	105.2	100.8	91.8
S1008	103.26	115.2	105.3	102.4
S1009	81.68	102.2	90.1	89.3
S1010	116.71	90.5	98.8	100.6
# W1000	ND	ND	ND	ND
W1001	96.26	122.2	104.1	161.4
W1003	74.22	81.7	102.2	90.9

μPADs: microfluidics paper-based analytical devices. UV–Vis: ultraviolet and visible spectrum colorimetric. CVAAS: cold vapor atomic absorption spectrum. ND: not detectable. Third-party assay: UNI-PRESIDENT ENTERPRISES CORP. Food Safety Center Food Safety Laboratory, methods: ICP-MS. * This test sample is listed in “2.1 Chemical Reagents”. The commercial products were purchased from Taiwan shopping malls, and the test results will be processed anonymously. # The W1000 sample was used to dissolve commercially available table salt and ddH2O with mercury standards.

## Data Availability

Not applicable.

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
