# Peer review of "Design of an Integrated Microfluidic Paper-Based Chip and Inspection Machine for the Detection of Mercury in Food with Silver Nanoparticles"

_biosensors, 2021, doi:10.3390/bios11120491_

Round 1
Reviewer 1 Report
As the title clearly explains, this manuscript is about the development of a paper-based analysis system for the detection of mercury in salt samples using silver nanoparticles and colorimetry. The authors describe the fabrication of the paper-based system, the development and calibration of the inspection machine, as well as experiments using commercial salt samples and lifetime evaluations. In the opinion of the reviewer, this is a very straightforward investigation, closer to development than scientific innovation. Some important comments/questions for the authors:
What is the novelty of this work? We have seen similar paper-based systems published repeatedly in the last decade or more. It was also mentioned in the manuscript that Meelapsom et al. performed a similar experiment for the application of μPAD for the determination of Hg2+ in water via a chromatic analysis based on a simple RGB color model and using AgNPs. If the technology has been developed by others and similar applications have been demonstrated (Hg2+ in water vs Hg in saltwater) what is the scientific contribution of this work?
Additionally, other smaller points need to be further addressed:
Line 63-64: quantitative and quantitative detections (please correct)
Line 81, 121, more places: refs
Figure 2A: what are the numbers on the x-axis?
Figure 3 A, B and C: results only shown for up to 5 minutes, not 6. Why?
Figure 3 D and E: the 1 ppb Hg samples were dropped. Why? The symbols also changed which makes it harder to compare to the first three plots of the figure.
Where did the Hg standard come from? Was it purchased by a company?
Were the salt samples anonymized for the experiment? Why?
In table 1, we see the range of results that the various measurement yielded. In some cases, the results show a large variation (e.g. S1006, W1001). No discussion was provided on this point. In some cases, the µPAD measurement will deem a sample unacceptable for consumption (over 100 ppb) when other measurements classify it as acceptable. What does that say about the possibilities of commercialization for this method?
In general, complete data sets need to be provided (see Figure 3) to facilitate comparison. When expected trends are not followed, proper explanations need to be provided. The simple data presentation provided needs to be followed by some critical discussion about the data.
Author Response
Comments and Suggestions for Authors
As the title clearly explains, this manuscript is about the development of a paper-based analysis system for the detection of mercury in salt samples using silver nanoparticles and colorimetry. The authors describe the fabrication of the paper-based system, the development and calibration of the inspection machine, as well as experiments using commercial salt samples and lifetime evaluations. In the opinion of the reviewer, this is a very straightforward investigation, closer to development than scientific innovation. Some important comments/questions for the authors:
Thanks to the reviewer's constructive suggestion to improve the quality of this manuscript, the author would give the highest respect.
What is the novelty of this work? We have seen similar paper-based systems published repeatedly in the last decade or more. It was also mentioned in the manuscript that Meelapsom et al. performed a similar experiment for the application of μPAD for the determination of Hg2+ in water via a chromatic analysis based on a simple RGB color model and using AgNPs. If the technology has been developed by others and similar applications have been demonstrated (Hg2+ in water vs Hg in saltwater) what is the scientific contribution of this work?
Response:
Thanks for your suggestions.
The scientific contributions of this work are:
- The volume ratio of color reagent: Hg standard (2:1) was optimal mix ratio for Hg content detect
- The R2 of the RGB values exceeded 0.99 (Hg concentration range 0.1-0.3 ppb; AgNP/ Hg volume ratio 2:1μL at 50 °C and 2.5 min; under a light source intensity of 2.7/2.7 V)
- µPADs are within the range of detection recovery rate of 70–120%.
- µPADs storage at low temperatures of 4–7 °C can use for approximately one month
- μPAD combined with AgNP can swift, economical, and simple detection method for Hg content in food.
The sentences have been rewritten. Line 507; Line 509-514.
Additionally, other smaller points need to be further addressed:
Line 63-64: quantitative and quantitative detections (please correct)
Response:
Thanks for your suggestions.
Corrected. Line 63-34.
Line 81, 121, more places: refs
Response:
Thanks for your suggestions.
Corrected. Line 81; Line 121; Line 129.
Figure 2A: what are the numbers on the x-axis?
Response:
Thanks for your suggestions.
Figure 2A is “Determining the best three primary color resolutions of standard color films under different light source intensities”, thus the x-axis is standard color chip intensity.
Figure 3 A, B and C: results only shown for up to 5 minutes, not 6. Why?
Response:
Thanks for your suggestions.
The Figure 3D, E have been redrawn.
Figure 3 D and E: the 1 ppb Hg samples were dropped. Why? The symbols also changed which makes it harder to compare to the first three plots of the figure.
Response:
Thanks for your suggestions.
The sentences have been rewritten. Line 407-411.
The Figure 3D, E have been redrawn.
Where did the Hg standard come from? Was it purchased by a company?
Response:
Thanks for your suggestions.
The sentences have been rewritten. Line 164-165.
Were the salt samples anonymized for the experiment? Why?
Response:
Thanks for your suggestions.
The sentences have been rewritten. Line 480-481.
In table 1, we see the range of results that the various measurement yielded. In some cases, the results show a large variation (e.g. S1006, W1001). No discussion was provided on this point. In some cases, the µPAD measurement will deem a sample unacceptable for consumption (over 100 ppb) when other measurements classify it as acceptable. What does that say about the possibilities of commercialization for this method?
Response:
Thanks for your suggestions.
The sentences have been rewritten. Line 446-452; Line 456-465; Line 467-471; Line 482.
In general, complete data sets need to be provided (see Figure 3) to facilitate comparison. When expected trends are not followed, proper explanations need to be provided. The simple data presentation provided needs to be followed by some critical discussion about the data.
Response:
Thanks for your suggestions.
The sentences have been rewritten.
Line 218-219; Line 274-276; Line 365-372; Line 393-399; Line 407-411; Line 413-421.
Chih-Yao Hou, Dr.
Department of Seafood Science, National Kaohsiung University of Science and Technology, Kaohsiung 811, Taiwan

Reviewer 2 Report
The authors presented in this manuscript a lateral flow sensor of mercury based on a colorimetric assay using silver nanoparticles. The proposed limit of detection is 0,1 ppb (I said proposed because it is not clear how it has been calculated) which is below the CVAAS technique which is the gold standard technique. They also studied the shelf-life of the prepared adsorbing pads by testing them after 28 days of storage. The manuscript could be of interest to the audience of this journal and to the field of mercury sensing. However, there are some issues that need to be addressed and I suggest to the authors to consider these comments/concerns.
1) The chemical formulas along the manuscript are not properly written. Hg2+ or Hg(0).
2) In Table S1 the volumes of the first two rows are not properly written.
3) The authors used two different reference styles.
4) I cannot observe any AgNPs in Figure 1. The authors do not mention the size of the AgNps obtained. Especially with Ag it is important because they can be very polydispersed with the mehotd used by the authors, and consequently affect the RGB variation and the chemical redox reaction. A TEM picture of the AgNPs should be provided (even if it is for the supplementary material).
5) It is not clear why 2.7/2.7 V is the best parameter. The R2 factor is very similar however there is a gap of 10. It is not clear what is the resolution gap.
6) The GAP definition was presented at page 14. It is should be presented at the beginning and not in the supplementary material.
7) It is not clear what is the minimum concentration of Hg that can be detected by this method. In table 1 are reported values around 100 ppb, however, at line 407 the range is 10 and 100 ppb.
8) How is it determined the limit of detection of this method? It is not explained.
Author Response
Comments and Suggestions for Authors
The authors presented in this manuscript a lateral flow sensor of mercury based on a colorimetric assay using silver nanoparticles. The proposed limit of detection is 0,1 ppb (I said proposed because it is not clear how it has been calculated) which is below the CVAAS technique which is the gold standard technique. They also studied the shelf-life of the prepared adsorbing pads by testing them after 28 days of storage. The manuscript could be of interest to the audience of this journal and to the field of mercury sensing. However, there are some issues that need to be addressed and I suggest to the authors to consider these comments/concerns.
Thanks to the reviewer's constructive suggestion to improve the quality of this manuscript, the author would give the highest respect.
1) The chemical formulas along the manuscript are not properly written. Hg2+ or Hg(0).
Response:
Thanks for your suggestions.
Corrected. Line 31, 37-39, 51, 53-54, 56-57, 113, 116, 119, 130-131, 136, 139, 147, 151-152, 234, 237, 240, 247, 257, 263, 267, 279, 287, 289-290, 298, 300, 304, 306, 324, 344, 350, 353, 360, 373, 383-385, 400, 404, 412, 427, 486, 496-497, 507-508, 518.
2) In Table S1 the volumes of the first two rows are not properly written.
Response:
Thanks for your suggestions.
Corrected. Line 556, Table S1.
3) The authors used two different reference styles.
Response:
Thanks for your suggestions.
Corrected. Line 81; Line 121; Line 129.
4) I cannot observe any AgNPs in Figure 1. The authors do not mention the size of the AgNps obtained. Especially with Ag it is important because they can be very polydispersed with the mehotd used by the authors, and consequently affect the RGB variation and the chemical redox reaction. A TEM picture of the AgNPs should be provided (even if it is for the supplementary material).
Response:
Thanks for your suggestions.
The Figure 1 has been redrawn.
The sentences have been rewritten. Line 317-319; Line 322-323.
5) It is not clear why 2.7/2.7 V is the best parameter. The R2 factor is very similar however there is a gap of 10. It is not clear what is the resolution gap.
Response:
Thanks for your suggestions.
The sentences have been rewritten. Line 206-208; Line 210-211; Line 336-339.
6) The GAP definition was presented at page 14. It is should be presented at the beginning and not in the supplementary material.
Response:
Thanks for your suggestions.
The sentences have been rewritten. Line 206-208; Line 210-211.
7) It is not clear what is the minimum concentration of Hg that can be detected by this method. In table 1 are reported values around 100 ppb, however, at line 407 the range is 10 and 100 ppb.
Response:
Thanks for your suggestions.
The sentences have been rewritten. Line 446-452; Line 467-471.
8) How is it determined the limit of detection of this method? It is not explained.
Response:
Thanks for your suggestions.
The sentences have been rewritten. Line274-276.
Chih-Yao Hou, Dr.
Department of Seafood Science, National Kaohsiung University of Science and Technology, Kaohsiung 811, Taiwan

Round 2
Reviewer 1 Report
Dear authors,
thank you for your adjustments to the manuscript. Some minor text editing is still required to correct spelling mistakes (e.g. parenthesis in line 463).
Author Response
Response to reviewer comments
thank you for your adjustments to the manuscript. Some minor text editing is still required to correct spelling mistakes (e.g. parenthesis in line 463).
Thanks again to the reviewer for the great guidance and insightful suggestions to improve the quality of this article, and the author would give the highest tribute.
Corrected. Line 471-472.
Chih-Yao Hou, Dr.
Department of Seafood Science, National Kaohsiung University of Science and Technology, Kaohsiung 811, Taiwan

Reviewer 2 Report
Dear Editor,
The authors replied to almost all comments raised by the reviewer, however, they did not answer to question number 8. Actually I believe that at line 274 there is a mistake since they wrote: "In this experiment, Refer to our previous research: the AgNP color.....". Clearly there is something wrong in this sentence. Furthermore, I can't find any definition of the LOD on these two new lines. I think it is important to define the LOD in this manuscript and not to add a self-citation.
What is the standard deviation of the AgNPs size?
Overall there are many typos along the manuscript that makes not possible to accept it in the present version. For instance line 408 has no clear meaning, as well as line 419-420. What is a portable orange devices? What is the meaning of line 464 with the sentence: "Measured mercury."?
Author Response
Response to reviewer comments
The authors replied to almost all comments raised by the reviewer, however, they did not answer to question number 8.
Thanks again to the reviewer for the great guidance and insightful suggestions to improve the quality of this article, and the author would give the highest tribute.
Actually I believe that at line 274 there is a mistake since they is something wrong in this sentence. Furthermore, I can't find any definition of the LOD on these two new lines. I think it is important to define the LOD in this manuscript and not to add a self-citation.
Response:
Thanks for your suggestions and apologizes for the negligence.
The sentences have been rewritten. Line 274; Line 293-297; Line 442-446.
What is the standard deviation of the AgNPs size?
Response:
Thanks for your suggestions.
The sentences have been rewritten. Line 321-325.
Figure 1 (A) has been redrawn.
The figure below is our previous data on AgNPs particle size distribution published in Micromachines (2021, https://doi.org/10.3390/mi12091123).
Figure 1(B). Nanoparticle Size Analyzer (NSA) size analysis of silver nanoparticles (AgNPs). The AgNP particle size was measured mainly at 47 nm.
Overall there are many typos along the manuscript that makes not possible to accept it in the present version. For instance line 408 has no clear meaning, as well as line 419-420. What is a portable orange devices? What is the meaning of line 464 with the sentence: "Measured mercury."?
Response:
Thanks for your suggestions and apologizes for the negligence.
The sentences have been rewritten. Line 413-415; Line 425-426; Line 471-472.
Chih-Yao Hou, Dr.
Department of Seafood Science, National Kaohsiung University of Science and Technology, Kaohsiung 811, Taiwan
